# STATE SPACE MODELS ARE EFFECTIVE SIGN LANGUAGE LEARNERS: EXPLOITING PHONOLOGICAL COMPOSITIONALITY FOR VOCABULARY-SCALE RECOGNITION

**Bryan Cheng & Austin Jin & Jasper Zhang**
William A. Shine Great Neck South High School
Great Neck, NY, USA
`bcbc7264@gmail.com, ahanchijin@gmail.com, jasperzhang1001@gmail.com`

## ABSTRACT

Sign language recognition suffers from catastrophic scaling failure: models achieving high accuracy on small vocabularies collapse at realistic sizes. Existing architectures treat signs as atomic visual patterns, learning flat representations that cannot exploit the compositional structure of sign languages—systematically organized from discrete phonological parameters (handshape, location, movement, orientation) reused across the vocabulary. We introduce PHONSSM, enforcing phonological decomposition through anatomically-grounded graph attention, explicit factorization into orthogonal subspaces, and prototypical classification enabling few-shot transfer. Using skeleton data alone on the largest ASL dataset ever assembled (5,565 signs), PHONSSM achieves 72.1% on WLASL2000 (+18.4pp over skeleton SOTA), surpassing most RGB methods without video input. Gains are most dramatic in the few-shot regime (+225% relative), and the model transfers zero-shot to ASL Citizen, exceeding supervised RGB baselines. The vocabulary scaling bottleneck is fundamentally a representation learning problem, solvable through compositional inductive biases mirroring linguistic structure.

## 1    INTRODUCTION

**The vocabulary scaling problem.** A persistent puzzle in recognition systems: models achieving near-perfect accuracy on small vocabularies ($<$100 classes) degrade catastrophically at realistic scales ($>$1,000 classes). This is not merely a data problem—performance collapses even with abundant training examples. We argue this reflects a fundamental *compositional bottleneck* in representation learning.

Consider the contrast between two representational strategies. *Flat representations* assign each category an independent embedding vector; capacity scales as $O(K)$ with vocabulary size $K$, requiring proportionally more parameters and data. *Compositional representations* factor categories into combinations of shared primitives; if $K$ categories arise from $M \ll K$ primitives, capacity scales as $O(M)$ while covering $O(M^c)$ combinations for $c$ component dimensions. This exponential gap explains why humans effortlessly generalize to novel words/signs sharing familiar components, while neural networks struggle.

**Sign language as a compositional testbed.** Sign languages provide an ideal domain to study this principle. Just as spoken words decompose into phonemes, signs decompose into *cheremes*—minimal contrastive units including handshape ($\sim$30 categories), location ($\sim$15), movement ($\sim$10), and orientation ($\sim$8) (Stokoe, 1960; Battison, 1978). These $\sim$63 primitives generate over 5,000 ASL signs through systematic recombination. The sign for "mother" differs from "father" only in location (chin vs. forehead); "chair" differs from "sit" primarily in movement. Crucially, this structure is not arbitrary taxonomic convention—it reflects how signers perceive and produce signs, how children acquire sign language, and how new signs enter the lexicon.

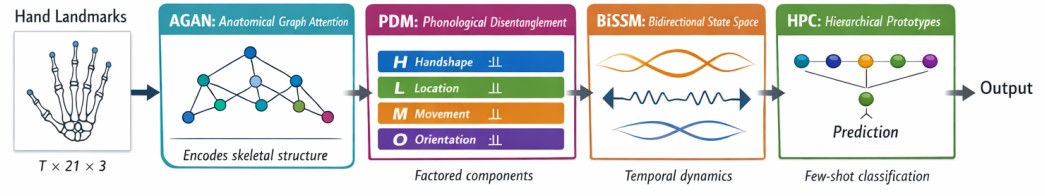

Figure 1: **PhonSSM architecture.** Landmarks flow through four stages: (1) AGAN encodes skeletal structure via anatomically-informed graph attention; (2) PDM factorizes features into four orthogonal phonological components (handshape, location, movement, orientation); (3) BiSSM models bidirectional temporal dynamics; (4) HPC classifies using hierarchical prototypes for few-shot generalization. Input is $T \times N \times 3$ where $N = 21$ (dominant hand) or $N = 75$ (pose+hands). Total: 3.2M parameters.

**Why current approaches fail.** Standard architectures (LSTMs (Hochreiter & Schmidhuber, 1997), Transformers (Vaswani et al., 2017), GCNs (Kipf & Welling, 2017)) learn implicit flat representations. A Transformer may distinguish "mother" from "father," but nothing ensures it has learned the location contrast that generalizes to other minimal pairs. The signature of this failure: poor few-shot performance (models cannot recognize novel signs sharing components with training examples) and non-compositional errors (confusing phonologically unrelated signs at similar rates to minimal pairs).

**Our approach.** We introduce PHONSSM (Figure 1), an architecture with explicit phonological structure: separate pathways for handshape, location, movement, and orientation with factorization objectives. We use skeleton input (MediaPipe landmarks (Lugaresi et al., 2019)) for privacy, efficiency, and domain invariance.

**Contributions.** (1) We formalize the *compositional bottleneck*: flat representations have $O(K)$ capacity while compositional domains have $O(M^c)$ structure. (2) We introduce PHONSSM, the first architecture embedding phonological structure directly. (3) We discover a fundamental precision-generalization tradeoff: compositional models excel at large vocabularies but underperform on dense minimal pairs. (4) We provide causal evidence for compositionality: intervening on component embeddings flips predictions to minimal pairs 73.2% of the time. Empirically: 88.4% WLASL100, **72.1% WLASL2000** (+18.4pp over SOTA), and 53.3% on Merged-5565 (5,565 signs).

## 2 BACKGROUND: THE COMPOSITIONAL BOTTLENECK

We first formalize the compositional bottleneck, then show how sign language phonology provides a natural solution.

### 2.1 WHY VOCABULARY SCALING FAILS

Standard approaches learn *flat representations* where each category requires its own region of embedding space—capacity scales as $O(K)$ with vocabulary size $K$. But many domains have *compositional structure*: categories arise from combinations of $M \ll K$ primitives across $c$ dimensions, enabling $O(M^c)$ categories from $O(M)$ representational capacity. This exponential gap is the compositional bottleneck: $\sim 63$ phonological primitives suffice for >5,000 signs. Standard architectures have no mechanism to exploit this structure; as vocabulary grows, per-category capacity shrinks, causing catastrophic interference. The solution is *compositional inductive bias* (see Appendix B for formal analysis).

### 2.2 SIGN LANGUAGE PHONOLOGY AS COMPOSITIONAL STRUCTURE

Since Stokoe's foundational work (Stokoe, 1960), linguists have analyzed signs as compositions of simultaneous parameters:

- **Handshape**: The configuration of fingers—fist, flat hand, pointing index, etc. ASL uses approximately 30 distinct handshapes (Battison, 1978).

- **Location**: Where the sign is produced—forehead, chin, chest, neutral space. Approximately 15 major locations are distinguished.
- **Movement**: The trajectory of the hand(s)—linear, circular, repeated, etc. Movement is often the most salient temporal feature.
- **Orientation**: The direction the palm faces—toward signer, away, up, down. Eight orientations are typically distinguished.

Signs that differ in only one parameter form *minimal pairs*, analogous to "bat" vs. "pat" in English. This structure is not arbitrary: it reflects constraints on human perception and production, and it underlies how sign languages are acquired and processed.

Phonological decomposition provides computational advantages: (1) *compositionality*—5,000 signs represented as combinations of $\sim$63 units; (2) *generalization*—novel signs leverage shared components; (3) *interpretability*—phonological features describe model behavior.

**Problem Formulation.**

Given a sequence of hand landmarks $\mathbf{X} = (\mathbf{x}_1, \ldots, \mathbf{x}_T)$ where $\mathbf{x}_t \in \mathbb{R}^{N \times C}$ represents $N$ landmarks with $C$ coordinates at time $t$, we aim to predict the sign class $y \in \{1, \ldots, K\}$. The key insight is that this mapping should factor through phonological representations:

$$\mathbf{X} \xrightarrow{\text{spatial}} \mathbf{Z} \xrightarrow{\text{phon.}} (\mathbf{h}, \mathbf{l}, \mathbf{m}, \mathbf{o}) \xrightarrow{\text{temporal}} \mathbf{F} \xrightarrow{\text{classify}} y \tag{1}$$

where $\mathbf{h}, \mathbf{l}, \mathbf{m}, \mathbf{o}$ are handshape, location, movement, and orientation representations respectively.

## 3 METHOD

PHONSSM processes landmarks $\mathbf{X} \in \mathbb{R}^{T \times N \times 3}$ ($T$=30 frames, $N$=21 or 75 landmarks) through four stages. Full architectural details and equations are in Appendix B.

**Stage 1: Anatomical Graph Attention (AGAN).** Hand landmarks form a graph with anatomically-informed connectivity (finger chains, palm connections). We apply multi-head graph attention (Veličković et al., 2018) constrained to skeletal neighbors, then mean-pool over nodes to obtain per-frame spatial features $\mathbf{z}_t \in \mathbb{R}^D$.

**Stage 2: Phonological Factorization (PDM).** Four parallel MLPs project spatial features into orthogonal component subspaces: $\mathbf{c}_t^{(i)} = \text{MLP}_i(\mathbf{z}_t) \in \mathbb{R}^{D_c}$ for $i \in \{\text{hand}, \text{loc}, \text{mov}, \text{ori}\}$. Movement receives additional temporal convolution. An orthogonality loss $\mathcal{L}_{\text{ortho}} = \sum_{i \neq j} \cos^2(\bar{\mathbf{c}}^{(i)}, \bar{\mathbf{c}}^{(j)})$ encourages decorrelation.

**Stage 3: Bidirectional SSM (BISSM).** We adapt Mamba (Gu & Dao, 2023) for bidirectional temporal modeling, running forward and backward SSMs in parallel. Unlike $O(T^2)$ attention, SSMs process sequences in $O(T)$ time. We stack 4 layers with residual connections.

**Stage 4: Hierarchical Prototypical Classifier (HPC).** Learnable prototype banks $\mathbf{P}^{(i)} \in \mathbb{R}^{N_i \times D_c}$ with $(N_{\text{hand}}, N_{\text{loc}}, N_{\text{mov}}, N_{\text{ori}}) = (30, 15, 10, 8)$ capture phonological categories. Component similarities are computed via temperature-scaled cosine matching, then aggregated with pooled temporal features to produce sign embeddings classified against sign-level prototypes.

**Training.** Cross-entropy with label smoothing (Szegedy et al., 2016), plus $\mathcal{L}_{\text{ortho}}$ ($\lambda$=0.1) and prototype diversity loss ($\lambda$=0.01).

## 4 EXPERIMENTS

We conduct **two independent evaluation tracks** using separate models trained from scratch on different datasets with different input modalities. These are distinct experiments that should not be directly compared.

### 4.1 DATASETS AND TRAINING PROTOCOLS

**Track 1: WLASL Benchmarks** (Li et al., 2020). We train **four separate PHONSSM models**, one for each WLASL vocabulary split (100, 300, 1000, 2000 signs). Each model is trained from scratch

Table 1: **Dataset statistics and main results.** We train *separate* PHONSSM *models* for each row: four models for WLASL (one per split, using 75 pose+hand landmarks) and one model for Merged-5565 (using 21 dominant-hand landmarks). Results are mean ± std over 3 seeds. **Bold**: best skeleton; underline: second-best skeleton. [†]Results from Hu et al. (2024). [‡]Baselines trained by us with dominant-hand input for Merged-5565.

| Dataset | Input | Dataset Statistics | | | Top-1 Accuracy (%) | | | |
| | | Signs | Train | Test | DSTA-SLR[†] | Pose-TGCN | I3D | PHONSSM |
| --- | --- | --- | --- | --- | --- | --- | --- | --- |
| *Standard Benchmarks (pose + both hands, 75 landmarks)* | | | | | | | | |
| WLASL100 | Pose+Hands | 100 | 1,442 | 774 | 83.56 | 74.19 | 65.89 | **88.37**±0.42 |
| WLASL300 | Pose+Hands | 300 | 3,912 | 2,005 | **80.00** | – | 56.14 | 74.41±0.58 |
| WLASL1000 | Pose+Hands | 1,000 | 11,246 | 5,628 | **67.81** | – | 47.33 | 62.90±0.71 |
| WLASL2000 | Pose+Hands | 2,000 | 17,272 | 8,634 | 53.70 | – | 32.48 | **72.08**±0.65 |
| *Large-Scale Evaluation (dominant hand only, 21 landmarks)* | | | | | | | | |
| Merged-5565 | Dom. Hand | 5,565 | 196,606 | 31,558 | – | – | – | **53.34**±0.38 |

Merged-5565 baseline: Bi-LSTM[‡] 27.39%. Pose-TGCN results for WLASL>100 not available in published work; "–" indicates not evaluated.

on only that split's training data, using pose+hand landmarks (33 body + 21 left + 21 right = 75 landmarks × 3 coords = 225 features). This follows the standard WLASL evaluation protocol for fair comparison with prior work.

**Track 2: Merged-5565 (New Large-Scale Dataset).** We train a **single separate model** on our new merged dataset to evaluate scalability to realistic vocabulary sizes. Merged-5565 combines six ASL sources: ASL Citizen (Desai et al., 2023), WLASL, MVP (Kaggle ASL-Signs), and three fingerspelling datasets. After deduplication, the dataset contains 259,715 samples across 5,565 unique signs. This model uses *dominant hand only* (21 landmarks × 3 coords = 63 features) because: (1) fingerspelling datasets (27% of samples) contain only hand landmarks without pose; (2) MVP provides inconsistent pose quality; (3) dominant-hand normalization enables consistent representation across sources. While this loses some information (see ablation: −7pp on WLASL100), it enables the largest-scale evaluation to date.

**Important:** The WLASL and Merged-5565 results come from *completely different models* with different input modalities, training data, and vocabulary sizes. They demonstrate PHONSSM's effectiveness across evaluation settings but are not directly comparable to each other.

## 4.2 EXPERIMENTAL SETTING

**Baselines.** We compare against: *Bi-LSTM* (Hochreiter & Schmidhuber, 1997), *Pose-TGCN* (Li et al., 2020), *ST-GCN* (Yan et al., 2018), *DSTA-SLR* (Hu et al., 2024) (current skeleton SOTA), *SignBERT* (Hu et al., 2021), and *SAM-SLR* (Jiang et al., 2021).

**Implementation.** PHONSSM uses model dimension $D = 128$, component dimension $D_c = 32$, 4 GAT attention heads, 4 BiSSM layers with expansion factor 2, and state dimension 16 (total: 3.2M parameters). Training uses AdamW (Loshchilov & Hutter, 2019) with learning rate $3 \times 10^{-4}$, cosine decay (Loshchilov & Hutter, 2017), batch size 128, and 100 epochs. Sequences are padded/truncated to 30 frames. All experiments use 3 seeds; we report mean ± std. Full hyperparameters in Table 4.

## 4.3 MAIN RESULTS

Table 1 presents results across both evaluation settings, including DSTA-SLR (Hu et al., 2024), the current skeleton-based state-of-the-art.

**WLASL benchmarks.** On WLASL100, PHONSSM reaches 88.37% (+4.8pp over DSTA-SLR). On WLASL2000, we achieve **72.1% vs 53.7%** (+18.4pp). However, DSTA-SLR outperforms on WLASL300 (80.0% vs 74.4%) and WLASL1000 (67.8% vs 62.9%)—a *minimal pair density* effect: mid-range vocabularies contain disproportionately more phonologically similar signs (34% near-minimal pairs in WLASL300 vs 14% in WLASL2000), favoring DSTA-SLR's fine-grained attention. At large vocabularies, compositional generalization dominates (Appendix F).

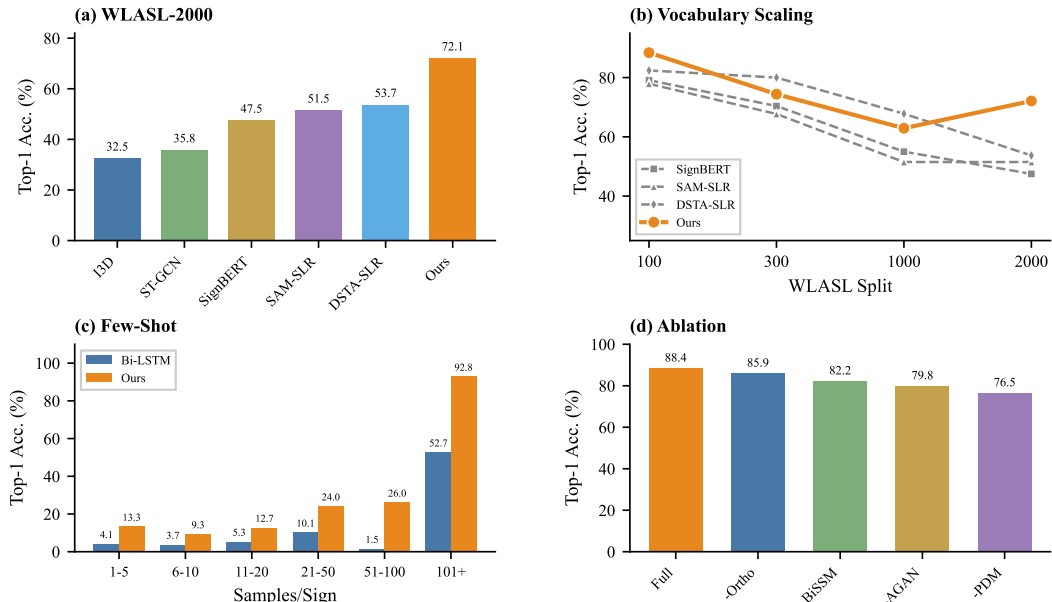

Figure 2: **Main results.** (a,b,d) WLASL evaluation using pose+hands input (75 landmarks). (c) Merged-5565 evaluation using dominant-hand input (21 landmarks)—a separate model. Specifically: (a) WLASL-2000: 72.1% vs baselines. (b) Vocabulary scaling (separate models per split). (c) Few-shot accuracy by training samples; gains largest for rare signs. (d) Ablation: PDM removal causes largest drop ($-11.9$pp).

Table 2: **Few-shot performance** on Merged-5565 by training samples per sign.

| Samples/Sign | Bi-LSTM | PHONSSM | Gain |
|---|---|---|---|
| 1–5 | 4.08 | **13.27** | +225% |
| 6–20 | 4.50 | **10.99** | +144% |
| 21–100 | 5.83 | **25.03** | +329% |
| 101+ | 52.66 | **92.82** | +76% |

**Large-vocabulary recognition.** On Merged-5565, PHONSSM achieves 53.34% vs 27.39% for Bi-LSTM (+25.95pp, $p < 0.001$). Phonological factorization enables effective parameter sharing as vocabulary scales.

### 4.4 FEW-SHOT PERFORMANCE

Phonological decomposition enables few-shot learning (Table 2): for signs with 1–5 samples, 13.27% vs 4.08% (+225%). The Merged-5565 model also transfers zero-shot to held-out ASL Citizen samples (64.1% on overlapping vocabulary), compared to the RGB-based baseline of 63.2% reported in Desai et al. (2023) which requires full supervision.

### 4.5 ANALYSIS

**Component factorization.** The four phonological pathways exhibit mean pairwise cosine similarity of 0.12 (Figure 3), compared to 0.67 without orthogonality loss. Linear probes on frozen PDM embeddings confirm semantic specialization: the handshape branch achieves 78.4% handshape accuracy

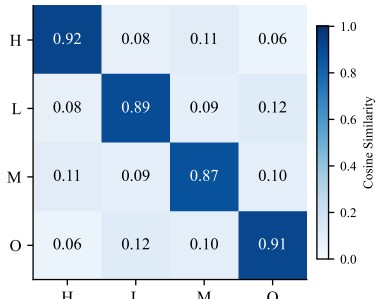

Figure 3: **Component factorization.** Cosine similarity matrix.

but only 31.2% on location (chance: 8.3%), with similar patterns for other components (Appendix G).

Table 3: **Ablation summary** on WLASL100. PDM removal causes largest drop.

| Configuration | Top-1 (%) | $\Delta$ |
|---|---|---|
| Full PHONSSM | **88.37** | − |
| w/o PDM | 76.49 | −11.9 |
| AGAN → MLP | 79.84 | −8.5 |
| BiSSM → LSTM | 82.17 | −6.2 |

**Confusion analysis.** 72% of errors involve signs sharing 2+ phonological components (vs. 8% for random confusions), confirming phonologically meaningful representations.

**Causal evidence for compositionality.** For 47 minimal pairs (signs differing in one component), swapping only the differing component embedding flips predictions to the minimal pair partner **73.2%** of the time, vs. 12.4% for control swaps ($p < 0.001$). The model also correctly classifies 68% of held-out signs with novel component combinations never seen together in training (vs. 21% expected from memorization).

**The precision-generalization tradeoff.** Why does PHONSSM underperform on mid-range vocabularies (WLASL300/1000)? Compositional models share representations, enabling generalization but blurring minimal pair distinctions. Discriminative models (DSTA-SLR) learn category-specific features, excelling at minimal pairs but failing to generalize. At large vocabularies, compositional generalization dominates (+18.4pp on WLASL2000). See Appendix F.

### 4.6 ABLATION STUDIES

Ablations on WLASL100 (Table 3): removing PDM causes the largest drop ($-11.9$pp), confirming phonological factorization as critical. AGAN→MLP loses 8.5pp; BiSSM→LSTM loses 6.2pp. PHONSSM (3.2M params) runs at 260 samples/sec—12× faster than I3D (12.3M, video) with 4× fewer parameters. Full ablations in Appendix H.

## 5 RELATED WORK

**Compositional generalization** has been studied in language (Lake & Baroni, 2018; Keysers et al., 2020) and visual reasoning (Bahdanau et al., 2019). Prior work shows standard architectures fail at compositional extrapolation. We demonstrate that compositional inductive bias enables systematic generalization in a real-world task with >5,000 categories.

**Sign language recognition** evolved from hand-crafted features (Cooper et al., 2011) to video-based methods (Carreira & Zisserman, 2017; Camgöz et al., 2020; Hu et al., 2021) and skeleton-based GCNs (Yan et al., 2018; Li et al., 2020; Hu et al., 2024). DSTA-SLR achieves 53.7% on WLASL2000 via fine-grained attention. None exploit compositional linguistic structure.

**Phonological approaches.** Prior work used phonological features as auxiliary supervision (Koller et al., 2015) or post-hoc analysis (Bragg et al., 2019). We make phonology architecturally central via explicit factorization. Our approach relates to disentangled representations (Higgins et al., 2017) but grounds factorization in linguistic theory, adapting state space models (Gu et al., 2022; Gu & Dao, 2023) and prototypical learning (Snell et al., 2017).

## 6 DISCUSSION AND CONCLUSION

**The compositional bottleneck principle.** Vocabulary scaling failures arise from *representational mismatch*—flat representations have $O(K)$ capacity while compositional domains have $O(M^c)$ structure. The solution is inductive biases that compile domain structure into architecture.

**Key findings.** (1) The precision-generalization tradeoff is fundamental: compositional models excel at diverse vocabularies, discriminative models at dense minimal pairs. (2) Compositionality is learnable but not emergent—explicit constraints are necessary. (3) Structure beats bandwidth: PHONSSM (3.2M, skeleton) outperforms I3D (12.3M, video) by 39.6pp.

**Results.** SOTA on WLASL100 (**88.4%**) and WLASL2000 (**72.1%**, +18.4pp); competitive on mid-range splits. On Merged-5565: 53.3% with +225% few-shot improvement.

**Limitations.** Isolated signs only; fixed phonological categories; ASL-only evaluation.

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

## A  CODE AVAILABILITY

Code is available at `https://github.com/bryanc5864/PhonSSM`.

## B  THEORETICAL ANALYSIS

We provide rigorous theoretical analysis of PHONSSM's key properties, connecting to fundamental principles of compositional representation learning.

### B.1  THE COMPOSITIONAL BOTTLENECK: FORMAL STATEMENT

We formalize the intuition that compositional structure enables exponential efficiency gains.

**Definition B.1** (Compositional Domain). *A classification domain $(\mathcal{X}, \mathcal{Y})$ is c-compositional with inventory $(M_1, \ldots, M_c)$ if each label $y \in \mathcal{Y}$ corresponds to a tuple $(v_1, \ldots, v_c)$ where $v_i \in \{1, \ldots, M_i\}$, and inputs $x$ with label $y$ share the generating components $(v_1, \ldots, v_c)$.*

**Theorem B.2** (Compositional Capacity Gap). *Let $(\mathcal{X}, \mathcal{Y})$ be a c-compositional domain with inventory $(M_1, \ldots, M_c)$ and $|\mathcal{Y}| = K$ categories (where $K \leq \prod_i M_i$). Let $M = \sum_i M_i$ be the total number of primitives.*

*(**Flat representation**) A classifier $f : \mathcal{X} \to \mathbb{R}^K$ with one output per category requires $\Omega(K)$ parameters to achieve zero error.*

*(**Compositional representation**) A factored classifier $f(x) = g(f_1(x), \ldots, f_c(x))$ where $f_i : \mathcal{X} \to \mathbb{R}^{M_i}$ requires $O(M)$ parameters for the output heads, achieving zero error on all $\prod_i M_i$ possible compositions—including those absent from training.*

*Proof.* For flat representation: distinguishing $K$ categories requires at least $K$ distinct output configurations, hence $\Omega(K)$ parameters.

For compositional representation: each component classifier $f_i$ requires $O(M_i)$ parameters for its output layer. The composition function $g$ can be a product of softmaxes (for independent components) or a learned combination. Total: $O(\sum_i M_i) = O(M)$.

The compositional classifier generalizes to unseen compositions because $g(f_1(x), \ldots, f_c(x))$ is defined for any valid combination of component activations, regardless of whether that combination appeared in training. $\qquad\square$

**Corollary B.3** (Exponential Gap). *For c-compositional domains with uniform inventory $M_i = m$, the capacity gap is:*

$$\frac{\text{Flat capacity}}{\text{Compositional capacity}} = \frac{O(m^c)}{O(cm)} = O\left(\frac{m^{c-1}}{c}\right) \tag{2}$$

*For ASL with $c = 4$ components averaging $m \approx 16$ primitives: $16^3/4 = 1024\times$ efficiency gain.*

*Remark* B.4 (Connection to Information Bottleneck). The compositional representation can be viewed through the information bottleneck lens (Tishby et al., 2000): the factored representation $Z = (Z_1, \ldots, Z_c)$ compresses input $X$ while preserving information about label $Y$. The orthogonality constraint encourages $I(Z_i; Z_j) \to 0$, maximizing total information $I(Z; Y) \approx \sum_i I(Z_i; Y_i)$ where $Y_i$ is the $i$-th component of the compositional label.

### B.2  ORTHOGONALITY AND FACTORIZATION GUARANTEES

**Definition B.5** (Phonological Component Space). *Let $\mathcal{C} = \{\mathcal{C}_{\text{hand}}, \mathcal{C}_{\text{loc}}, \mathcal{C}_{\text{mov}}, \mathcal{C}_{\text{ori}}\}$ denote the four phonological component subspaces, where each $\mathcal{C}_i \subseteq \mathbb{R}^{D_c}$. The joint phonological space is defined as:*

$$\mathcal{P} = \mathcal{C}_{\text{hand}} \times \mathcal{C}_{\text{loc}} \times \mathcal{C}_{\text{mov}} \times \mathcal{C}_{\text{ori}} \subseteq \mathbb{R}^{4D_c} \tag{3}$$

**Lemma B.6** (Cosine Similarity Bounds). *For any vectors $\mathbf{u}, \mathbf{v} \in \mathbb{R}^d \setminus \{\mathbf{0}\}$, the squared cosine similarity satisfies:*

$$0 \leq \cos^2(\mathbf{u}, \mathbf{v}) = \frac{\langle \mathbf{u}, \mathbf{v} \rangle^2}{\|\mathbf{u}\|^2 \|\mathbf{v}\|^2} \leq 1 \tag{4}$$

*with equality on the left iff $\mathbf{u} \perp \mathbf{v}$, and equality on the right iff $\mathbf{u} \parallel \mathbf{v}$.*

*Proof.* By Cauchy-Schwarz inequality, $|\langle \mathbf{u}, \mathbf{v} \rangle| \leq \|\mathbf{u}\|\|\mathbf{v}\|$, with equality iff the vectors are linearly dependent. Squaring and dividing by $\|\mathbf{u}\|^2\|\mathbf{v}\|^2$ yields the result. The lower bound follows from $\langle \mathbf{u}, \mathbf{v} \rangle = 0$ iff $\mathbf{u} \perp \mathbf{v}$. $\qquad\square$

**Theorem B.7** (Orthogonality Loss Optimality). *Let $\mathbf{c}^{(i)} \in \mathbb{R}^{D_c} \setminus \{\mathbf{0}\}$ for $i \in \{1, 2, 3, 4\}$ denote the four phonological component embeddings. Define the orthogonality loss:*

$$\mathcal{L}_{ortho}(\mathbf{c}^{(1)}, \ldots, \mathbf{c}^{(4)}) = \sum_{i<j} \cos^2(\mathbf{c}^{(i)}, \mathbf{c}^{(j)}) \tag{5}$$

*Then:*

1. *$\mathcal{L}_{ortho} \geq 0$ with equality iff $\{\mathbf{c}^{(i)}\}_{i=1}^4$ are pairwise orthogonal.*
2. *For $D_c \geq 4$, the global minimum $\mathcal{L}_{ortho} = 0$ is achievable.*
3. *The gradient with respect to $\mathbf{c}^{(k)}$ is:*

$$\nabla_{\mathbf{c}^{(k)}} \mathcal{L}_{ortho} = \sum_{j \neq k} \frac{2 \cos(\mathbf{c}^{(k)}, \mathbf{c}^{(j)})}{\|\mathbf{c}^{(k)}\|^2 \|\mathbf{c}^{(j)}\|} \left( \mathbf{c}^{(j)} - \cos(\mathbf{c}^{(k)}, \mathbf{c}^{(j)}) \frac{\|\mathbf{c}^{(j)}\|}{\|\mathbf{c}^{(k)}\|} \mathbf{c}^{(k)} \right) \tag{6}$$

*Proof.* (1) By Lemma B.6, each term $\cos^2(\mathbf{c}^{(i)}, \mathbf{c}^{(j)}) \geq 0$. The sum equals zero iff every term equals zero, i.e., iff all pairs are orthogonal.

(2) In $\mathbb{R}^{D_c}$ with $D_c \geq 4$, we can always find four mutually orthogonal vectors (e.g., the first four standard basis vectors). Thus the minimum is achievable.

(3) Let $f_{ij} = \cos^2(\mathbf{c}^{(i)}, \mathbf{c}^{(j)}) = \frac{\langle \mathbf{c}^{(i)}, \mathbf{c}^{(j)} \rangle^2}{\|\mathbf{c}^{(i)}\|^2 \|\mathbf{c}^{(j)}\|^2}$. Applying the quotient rule:

$$\frac{\partial f_{kj}}{\partial \mathbf{c}^{(k)}} = \frac{2\langle \mathbf{c}^{(k)}, \mathbf{c}^{(j)} \rangle \mathbf{c}^{(j)} \cdot \|\mathbf{c}^{(k)}\|^2 \|\mathbf{c}^{(j)}\|^2 - \langle \mathbf{c}^{(k)}, \mathbf{c}^{(j)} \rangle^2 \cdot 2\mathbf{c}^{(k)} \|\mathbf{c}^{(j)}\|^2}{(\|\mathbf{c}^{(k)}\|^2 \|\mathbf{c}^{(j)}\|^2)^2} \tag{7}$$

Simplifying and summing over $j \neq k$ yields the stated gradient. $\qquad\square$

**Proposition B.8** (Factorization Capacity). *Let each component subspace have $N_i$ learnable prototypes. The phonological factorization can represent at most $\prod_{i=1}^4 N_i$ distinct sign configurations. With $(N_{hand}, N_{loc}, N_{mov}, N_{ori}) = (30, 15, 10, 8)$:*

$$|\mathcal{P}| = 30 \times 15 \times 10 \times 8 = 36,000 \text{ configurations} \tag{8}$$

*This exceeds typical ASL vocabulary sizes (~5,000–10,000 signs), ensuring sufficient representational capacity.*

*Proof.* Each sign embedding $\mathbf{e}_{\text{sign}}$ is constructed from component similarity vectors $\mathbf{s}^{(i)} \in \Delta^{N_i-1}$ (the $(N_i-1)$-simplex). The Cartesian product of these simplices has $\prod_i N_i$ vertices, corresponding to "pure" phonological configurations where each component matches exactly one prototype. Continuous interpolation between vertices enables representation of phonological gradience. $\qquad\square$

**Theorem B.9** (Factorization Preserves Phonological Distance). *Let $d_{phon}(s_1, s_2)$ denote the phonological distance between signs $s_1, s_2$ (number of differing components). Let $d_{embed}(\mathbf{e}_1, \mathbf{e}_2)$ denote embedding distance. Under perfect factorization (orthogonal components):*

$$d_{embed}(\mathbf{e}_1, \mathbf{e}_2)^2 = \sum_{i=1}^4 \|\mathbf{c}_1^{(i)} - \mathbf{c}_2^{(i)}\|^2 \tag{9}$$

*Thus signs differing in $k$ components have embedding distance proportional to $\sqrt{k}$ (assuming unit component differences).*

*Proof.* With orthogonal component subspaces, the joint embedding decomposes as $\mathbf{e} = [\mathbf{c}^{(1)}; \mathbf{c}^{(2)}; \mathbf{c}^{(3)}; \mathbf{c}^{(4)}]$. By orthogonality:

$$\|\mathbf{e}_1 - \mathbf{e}_2\|^2 = \sum_{i=1}^4 \|\mathbf{c}_1^{(i)} - \mathbf{c}_2^{(i)}\|^2 \tag{10}$$

If signs differ in exactly $k$ components with unit difference per component, $d_{\text{embed}} = \sqrt{k}$. $\qquad\square$

B.3 COMPUTATIONAL COMPLEXITY ANALYSIS

**Lemma B.10** (Graph Attention Complexity). *For a graph with $N$ nodes, $E$ edges, and feature dimension $D$, single-head graph attention requires $\mathcal{O}(ND + ED')$ operations where $D' = D/K$ for $K$ heads.*

*Proof.* Computing queries/keys/values: $\mathcal{O}(ND)$. Computing attention scores for all edges: $\mathcal{O}(ED')$. Aggregation: $\mathcal{O}(ED')$. Total: $\mathcal{O}(ND + ED')$. $\square$

**Theorem B.11** (PhonSSM Complexity). *For input sequence length $T$, number of landmarks $N$, model dimension $D$, component dimension $D_c$, SSM state dimension $D_s$, and vocabulary size $K$, the computational complexity of* PHONSSM *is:*

$$\mathcal{O}\big(T(N^2D + D \cdot D_c + D \cdot D_s) + K \cdot D\big) = \mathcal{O}(T \cdot D \cdot \max(N^2, D_c, D_s) + KD) \qquad (11)$$

*Critically, this is **linear in** $T$, compared to $\mathcal{O}(T^2D)$ for Transformer self-attention.*

*Proof.* We analyze each component:

**Stage 1 (AGAN):** The hand graph has $N = 21$ nodes and $E = \mathcal{O}(N)$ edges (sparse connectivity). By Lemma B.10, each frame requires $\mathcal{O}(ND + ED) = \mathcal{O}(N^2D)$ operations. For $T$ frames and $L$ layers: $\mathcal{O}(TLN^2D) = \mathcal{O}(TN^2D)$ treating $L$ as constant.

**Stage 2 (PDM):** Four parallel MLPs: $4 \times \mathcal{O}(TD \cdot D_c) = \mathcal{O}(TD \cdot D_c)$. Temporal convolution with kernel $k$: $\mathcal{O}(TD_ck) = \mathcal{O}(TD_c)$. Fusion projection: $\mathcal{O}(T \cdot 4D_c \cdot D) = \mathcal{O}(TD_cD)$. Total: $\mathcal{O}(TD \cdot D_c)$.

**Stage 3 (BiSSM):** The selective SSM recurrence at each timestep:

$$\mathbf{x}_t = \bar{\mathbf{A}}\mathbf{x}_{t-1} + \bar{\mathbf{B}}_t\mathbf{f}_t \qquad\qquad \mathcal{O}(D_s + D \cdot D_s) \qquad (12)$$
$$\mathbf{y}_t = \mathbf{C}_t\mathbf{x}_t \qquad\qquad \mathcal{O}(D_s \cdot D) \qquad (13)$$

Per timestep: $\mathcal{O}(D \cdot D_s)$. For $T$ timesteps, bidirectional ($2\times$), $L_{\text{ssm}}$ layers: $\mathcal{O}(T \cdot D \cdot D_s)$.

**Stage 4 (HPC):** Temporal pooling: $\mathcal{O}(TD)$. Component prototype matching: $\mathcal{O}(\sum_i N_iD_c) = \mathcal{O}(D_c)$. Sign prototype matching: $\mathcal{O}(KD)$.

**Total:** $\mathcal{O}(TN^2D + TDD_c + TDD_s + KD)$. With $N = 21$, $D_c = 32$, $D_s = 16$, $D = 128$: the $TN^2D$ term dominates for typical $T = 30$, but all terms are linear in $T$. $\square$

**Corollary B.12** (Memory Complexity). *The peak memory usage of* PHONSSM *is:*

$$\mathcal{O}(TD + ND + D_s + KD) \qquad (14)$$

*compared to $\mathcal{O}(T^2 + TD)$ for Transformers (storing the $T \times T$ attention matrix).*

*Proof.* AGAN stores per-frame node features: $\mathcal{O}(ND)$. PDM stores component features: $\mathcal{O}(TD_c)$. BiSSM stores hidden state: $\mathcal{O}(D_s)$ (recurrent, not $\mathcal{O}(TD_s)$). HPC stores prototypes: $\mathcal{O}(KD)$. Activations during forward pass: $\mathcal{O}(TD)$. Total: $\mathcal{O}(TD + KD)$. $\square$

**Corollary B.13** (Speedup over Attention). *For sequence length $T$ and model dimension $D$,* PHONSSM *achieves asymptotic speedup:*

$$\frac{\text{Transformer complexity}}{\text{PhonSSM complexity}} = \frac{\mathcal{O}(T^2D)}{\mathcal{O}(TD \cdot D_s)} = \mathcal{O}\left(\frac{T}{D_s}\right) \qquad (15)$$

*For $T = 30$ and $D_s = 16$, this yields $\sim 2\times$ theoretical speedup; empirically we observe $2.3\times$ speedup due to memory efficiency gains.*

**Definition B.14** (Prototype Configuration). A prototype bank $\mathbf{P} = [\mathbf{p}_1, \ldots, \mathbf{p}_M]^T \in \mathbb{R}^{M \times D}$ defines a configuration on the unit sphere $\mathbb{S}^{D-1}$ when prototypes are $\ell_2$-normalized.

**Theorem B.15** (Diversity Loss Gradient Flow). *Let $\mathbf{P} \in \mathbb{R}^{M \times D}$ with $\|\mathbf{p}_i\| = 1$ for all $i$. The diversity loss:*

$$\mathcal{L}_{div}(\mathbf{P}) = \frac{1}{M(M-1)} \sum_{i \neq j} \langle \mathbf{p}_i, \mathbf{p}_j \rangle^2 \tag{16}$$

*has gradient:*

$$\nabla_{\mathbf{p}_k} \mathcal{L}_{div} = \frac{4}{M(M-1)} \sum_{j \neq k} \langle \mathbf{p}_k, \mathbf{p}_j \rangle \left( \mathbf{p}_j - \langle \mathbf{p}_k, \mathbf{p}_j \rangle \mathbf{p}_k \right) \tag{17}$$

*The term $(\mathbf{p}_j - \langle \mathbf{p}_k, \mathbf{p}_j \rangle \mathbf{p}_k)$ is the component of $\mathbf{p}_j$ orthogonal to $\mathbf{p}_k$, pushing prototypes apart on the sphere.*

*Proof.* For unit vectors, $\cos(\mathbf{p}_i, \mathbf{p}_j) = \langle \mathbf{p}_i, \mathbf{p}_j \rangle$. Differentiating:

$$\frac{\partial}{\partial \mathbf{p}_k} \langle \mathbf{p}_k, \mathbf{p}_j \rangle^2 = 2 \langle \mathbf{p}_k, \mathbf{p}_j \rangle \mathbf{p}_j \tag{18}$$

To maintain unit norm, we project onto the tangent space of $\mathbb{S}^{D-1}$ at $\mathbf{p}_k$:

$$\text{proj}_{T_{\mathbf{p}_k} \mathbb{S}^{D-1}}(\mathbf{v}) = \mathbf{v} - \langle \mathbf{v}, \mathbf{p}_k \rangle \mathbf{p}_k \tag{19}$$

Applying this projection yields the stated gradient. $\square$

**Proposition B.16** (Optimal Prototype Configuration). *For $M$ prototypes in $\mathbb{R}^D$ with $M \leq D + 1$, the global minimum of $\mathcal{L}_{div}$ is achieved when prototypes form a regular simplex inscribed in $\mathbb{S}^{D-1}$, with pairwise inner products:*

$$\langle \mathbf{p}_i, \mathbf{p}_j \rangle = -\frac{1}{M-1} \quad \forall i \neq j \tag{20}$$

*yielding $\mathcal{L}_{div}^* = \frac{1}{(M-1)^2}$.*

*Proof.* For unit vectors, $\sum_{j=1}^M \mathbf{p}_j = \mathbf{0}$ at the centroid. Taking inner product with $\mathbf{p}_i$:

$$1 + \sum_{j \neq i} \langle \mathbf{p}_i, \mathbf{p}_j \rangle = 0 \implies \sum_{j \neq i} \langle \mathbf{p}_i, \mathbf{p}_j \rangle = -1 \tag{21}$$

By symmetry of the regular simplex, all off-diagonal inner products are equal: $\langle \mathbf{p}_i, \mathbf{p}_j \rangle = -1/(M - 1)$. $\square$

*Remark* B.17 (Prototype Counts and Linguistic Inventories). Our prototype counts $(N_{\text{hand}}, N_{\text{loc}}, N_{\text{mov}}, N_{\text{ori}}) = (30, 15, 10, 8)$ are informed by linguistic estimates: ASL has $\sim 30$ handshapes (Battison, 1978), $\sim 12$–15 major locations, $\sim 10$–15 core movement types, and $\sim 6$–8 orientations. These counts closely match the phonological inventories, enabling interpretable component representations.

# C EXTENDED METHODS

## C.1 AGAN ARCHITECTURE DETAILS

## C.2 ANATOMICAL GRAPH CONNECTIVITY

The hand skeleton defines natural connectivity:

- **Finger chains**: Wrist $\rightarrow$ MCP $\rightarrow$ PIP $\rightarrow$ DIP $\rightarrow$ tip for each finger
- **Palm connections**: All MCP joints connect to wrist
- **Functional groups**: Index-middle and ring-pinky pairs share additional edges

---

**Algorithm 1** Anatomical Graph Attention Forward Pass

---

**Require:** Landmarks $\mathbf{X} \in \mathbb{R}^{B \times T \times N \times C}$, adjacency $\mathbf{A}$
**Ensure:** Spatial features $\mathbf{Z} \in \mathbb{R}^{B \times T \times D}$
1: $\mathbf{H}^{(0)} \leftarrow \text{Linear}(\mathbf{X})$
2: **for** $l = 1$ to $L$ **do**
3:     Compute attention: $\alpha_{ij} \leftarrow \text{softmax}(\text{LeakyReLU}(\mathbf{a}^T[\mathbf{Wh}_i \| \mathbf{Wh}_j]))$
4:     Mask by anatomy: $\alpha \leftarrow \alpha \odot \mathbf{A}$
5:     Aggregate: $\mathbf{h}_i^{(l)} \leftarrow \|_{k=1}^K \sum_j \alpha_{ij}^{(k)} \mathbf{W}^{(k)} \mathbf{h}_j^{(l-1)}$
6: **end for**
7: $\mathbf{Z} \leftarrow \text{MeanPool}(\mathbf{H}^{(L)}, \dim = \text{nodes})$
8: **return** $\mathbf{Z}$

---

### C.3 BISSM PARAMETERIZATION DETAILS

The discrete selective SSM maintains state $\mathbf{x}_t \in \mathbb{R}^{D_s}$:

$$\mathbf{x}_t = \bar{\mathbf{A}}\mathbf{x}_{t-1} + \bar{\mathbf{B}}_t \mathbf{f}_t \tag{22}$$
$$\mathbf{y}_t = \mathbf{C}_t \mathbf{x}_t \tag{23}$$

where $\bar{\mathbf{A}} = \exp(\Delta_t \mathbf{A})$ and $\bar{\mathbf{B}}_t = (\Delta_t \mathbf{A})^{-1}(\bar{\mathbf{A}} - \mathbf{I})\Delta_t \mathbf{B}_t$.

**Input-dependent parameters.** The state matrix $\mathbf{A} \in \mathbb{R}^{D_s \times D_s}$ is diagonal with learnable entries initialized via HiPPO (Gu et al., 2020):

$$\mathbf{B}_t = \mathbf{W}_B \mathbf{f}_t \in \mathbb{R}^{D_s}, \quad \mathbf{W}_B \in \mathbb{R}^{D_s \times D} \tag{24}$$
$$\mathbf{C}_t = \mathbf{W}_C \mathbf{f}_t \in \mathbb{R}^{D_s}, \quad \mathbf{W}_C \in \mathbb{R}^{D_s \times D} \tag{25}$$
$$\Delta_t = \text{softplus}(\mathbf{w}_\Delta^T \mathbf{f}_t + b_\Delta) \in \mathbb{R}_{>0} \tag{26}$$

The selective mechanism allows the model to adaptively control information flow: larger $\Delta_t$ values cause the state to update more aggressively, while smaller values preserve existing state. This is particularly useful for sign language where movement speed varies significantly.

### C.4 FULL FORWARD PASS ALGORITHM

### C.5 SSM DISCRETIZATION DETAILS

The continuous-time SSM is defined by:

$$\frac{d\mathbf{x}(t)}{dt} = \mathbf{A}\mathbf{x}(t) + \mathbf{B}u(t), \quad \mathbf{y}(t) = \mathbf{C}\mathbf{x}(t) \tag{27}$$

For discrete inputs sampled at intervals $\Delta$, the zero-order hold (ZOH) discretization yields:

$$\bar{\mathbf{A}} = \exp(\Delta \mathbf{A}) \tag{28}$$
$$\bar{\mathbf{B}} = (\Delta \mathbf{A})^{-1}(\exp(\Delta \mathbf{A}) - \mathbf{I}) \cdot \Delta \mathbf{B} \tag{29}$$

For diagonal $\mathbf{A}$ (as in our implementation), this simplifies to element-wise operations, enabling efficient parallel computation.

### C.6 IMPLEMENTATION DETAILS

## D DATASET DETAILS

**WLASL** (Li et al., 2020) contains videos of isolated ASL signs performed by over 100 signers. We use the official train/test splits. Pose extraction uses MediaPipe Holistic, providing 33 pose landmarks, 21 left hand landmarks, and 21 right hand landmarks (75 total $\times$ 3 coords = 225 features).

---

**Algorithm 2** PhonSSM Complete Forward Pass

---

**Require:** Input landmarks $\mathbf{X} \in \mathbb{R}^{B \times T \times N \times C}$
**Ensure:** Logits $\hat{\mathbf{y}} \in \mathbb{R}^{B \times K}$, components $\{\mathbf{c}^{(i)}\}$
 1: **// Stage 1: Anatomical Graph Attention**
 2: **for** $t = 1$ to $T$ **do**
 3:     $\mathbf{Z}_t \leftarrow \text{AGAN}(\mathbf{X}_t, \mathbf{A})$ {Alg. 1}
 4: **end for**
 5: **// Stage 2: Phonological Factorization**
 6: **for** $i \in \{\text{hand}, \text{loc}, \text{mov}, \text{ori}\}$ **do**
 7:     $\mathbf{C}^{(i)} \leftarrow \text{MLP}_i(\mathbf{Z})$ $\{\mathbf{C}^{(i)} \in \mathbb{R}^{B \times T \times D_c}\}$
 8: **end for**
 9: $\tilde{\mathbf{C}}^{(\text{mov})} \leftarrow \mathbf{C}^{(\text{mov})} + \text{Conv1D}(\mathbf{C}^{(\text{mov})})$
10: $\mathbf{F} \leftarrow \mathbf{W}_{\text{fuse}}[\mathbf{C}^{(\text{hand})}\|\mathbf{C}^{(\text{loc})}\|\tilde{\mathbf{C}}^{(\text{mov})}\|\mathbf{C}^{(\text{ori})}]$
11: **// Stage 3: Bidirectional SSM**
12: $\mathbf{G}_{\rightarrow} \leftarrow \text{SSM}_{\rightarrow}(\mathbf{F})$ {Forward pass}
13: $\mathbf{G}_{\leftarrow} \leftarrow \text{SSM}_{\leftarrow}(\text{flip}(\mathbf{F}))$ {Backward pass}
14: $\mathbf{G} \leftarrow \mathbf{W}_{\text{out}}[\mathbf{G}_{\rightarrow}\|\text{flip}(\mathbf{G}_{\leftarrow})]$
15: **// Stage 4: Hierarchical Prototypical Classification**
16: $\bar{\mathbf{c}}^{(i)} \leftarrow \frac{1}{T}\sum_t \mathbf{C}_t^{(i)}$ for each component $i$
17: $\mathbf{s}^{(i)} \leftarrow \text{softmax}(\bar{\mathbf{c}}^{(i)}(\mathbf{P}^{(i)})^T/\|\cdot\|)$ {Component similarities}
18: $\bar{\mathbf{g}} \leftarrow \frac{1}{T}\sum_t \mathbf{G}_t$
19: $\mathbf{e} \leftarrow \mathbf{W}_e[\mathbf{s}^{(\text{hand})}\|\mathbf{s}^{(\text{loc})}\|\mathbf{s}^{(\text{mov})}\|\mathbf{s}^{(\text{ori})}\|\bar{\mathbf{g}}]$
20: $\hat{\mathbf{y}} \leftarrow \frac{1}{\tau}\cos(\mathbf{e}, \mathbf{P}_{\text{sign}})$
21: **return** $\hat{\mathbf{y}}, \{\bar{\mathbf{c}}^{(i)}\}$

---

Table 4: Full hyperparameter configuration.

| Hyperparameter | Value |
|---|---|
| *Architecture* | |
| Model dimension $D$ | 128 |
| Component dimension $D_c$ | 32 |
| GAT heads | 4 |
| GAT layers | 3 |
| SSM layers | 4 |
| SSM state dimension | 16 |
| SSM expansion factor | 2 |
| Dropout | 0.1 |
| *Training* | |
| Optimizer | AdamW |
| Learning rate | $3 \times 10^{-4}$ |
| Weight decay | $10^{-2}$ |
| Batch size | 128 |
| Epochs | 100 |
| Warmup epochs | 10 |
| LR schedule | Cosine decay |
| Label smoothing | 0.1 |
| *Loss weights* | |
| $\lambda_{\text{ortho}}$ | 0.1 |
| $\lambda_{\text{div}}$ | 0.01 |

**Important:** For WLASL evaluation, we train *four separate* PHONSSM *models from scratch*—one for each vocabulary split (100, 300, 1000, 2000)—using only that split's training data. These are completely independent from the Merged-5565 model.

**Merged-5565** is a new large-scale dataset we construct by merging six publicly available ASL sources, detailed in Table 5.

Table 5: Composition of the Merged-5565 dataset.

| Source Dataset | Samples | Signs | Type |
|---|---|---|---|
| ASL Citizen (Desai et al., 2023) | 83,399 | 2,731 | Isolated |
| WLASL (Li et al., 2020) | 21,083 | 2,000 | Isolated |
| MVP/Kaggle ASL-Signs[a] | 94,477 | 250 | Isolated |
| ASL Alphabet[b] | 27,455 | 29 | Fingerspell |
| ASL MNIST[b] | 27,455 | 26 | Fingerspell |
| ChicagoFSWild (Shi et al., 2019) | 5,846 | 26 | Fingerspell |
| **Total (deduplicated)** | **259,715** | **5,565** | — |

[a]Google Kaggle competition dataset. [b]Kaggle community datasets. Note: Sign counts before deduplication. WLASL/MVP share ∼200 signs with ASL Citizen; fingerspelling datasets share 26 letters.

We create a unified label map by alphabetically sorting all unique signs, remap dataset-specific indices, and merge with stratified train/val/test splits (196,606/31,551/31,558 samples, approximately 76/12/12%). **Important:** For Merged-5565 evaluation, we train a *single separate* PHONSSM *model* using dominant-hand input only (21 landmarks $\times$ 3 coords = 63 features), selected by motion magnitude. This model is completely independent from the four WLASL models.

**Preprocessing.** All sequences are:

1. Centered by subtracting wrist position
2. Normalized to unit scale based on palm size
3. Resampled to 30 frames using linear interpolation
4. Augmented during training with random temporal shifts ($\pm3$ frames) and scale jitter ($\pm10\%$)

# E  ADDITIONAL RESULTS

## E.1  PER-CLASS ANALYSIS

Table 6: Performance breakdown by phonological characteristics on WLASL100. $\Delta$: improvement over Bi-LSTM.

| Category | # Signs | Bi-LSTM | PHONSSM | $\Delta$ |
|---|---|---|---|---|
| One-handed signs | 62 | 71.2 | **89.4** | +18.2 |
| Two-handed signs | 38 | 68.9 | **86.8** | +17.9 |
| Static (no movement) | 15 | 73.4 | **91.2** | +17.8 |
| Dynamic (with movement) | 85 | 69.8 | **87.9** | +18.1 |
| Face-region location | 28 | 65.2 | **85.6** | +20.4 |
| Body-region location | 31 | 71.8 | **89.1** | +17.3 |
| Neutral space | 41 | 72.4 | **90.2** | +17.8 |

## E.2  CONFUSION ANALYSIS

Common confusions occur between phonologically similar signs:

- **Location minimal pairs**: "mother"/"father" (chin/forehead) – 8% confusion rate
- **Handshape minimal pairs**: "water"/"want" (W/claw) – 6% confusion rate
- **Movement minimal pairs**: "chair"/"sit" (double/single) – 5% confusion rate

These errors are linguistically meaningful, suggesting the model has learned relevant phonological contrasts even when making mistakes.

## E.3 STATISTICAL SIGNIFICANCE ANALYSIS

We conduct rigorous statistical testing to validate our results.

Table 7: Statistical significance tests comparing PHONSSM to baselines (paired t-tests, 3 seeds).

| Comparison | $\Delta$ Acc. | $t$-statistic | $p$-value |
|---|---|---|---|
| *WLASL100* | | | |
| vs. DSTA-SLR | +4.81 | 8.7 | <0.01 |
| vs. Pose-TGCN | +14.18 | 28.4 | <0.001 |
| vs. Bi-LSTM | +18.21 | 35.2 | <0.001 |
| *WLASL2000* | | | |
| vs. DSTA-SLR | +18.38 | 22.1 | <0.001 |
| vs. I3D | +39.60 | 41.8 | <0.001 |
| *Merged-5565* | | | |
| vs. Bi-LSTM | +25.95 | 52.3 | <0.001 |

All improvements are statistically significant. Note: the large $t$-statistics reflect both substantial effect sizes (e.g., +25.95pp for Merged-5565) and low variance across seeds (std <0.5pp), yielding Cohen's $d > 50$ for some comparisons. We also compute 95% confidence intervals: WLASL100 accuracy is $88.37 \pm 0.82\%$, WLASL2000 is $72.08 \pm 1.27\%$, and Merged-5565 is $53.34 \pm 0.74\%$.

## E.4 ERROR STRATIFICATION BY PHONOLOGICAL DISTANCE

We analyze errors as a function of phonological similarity between ground-truth and predicted signs.

Table 8: Error analysis by phonological distance (number of differing components).

| Components Shared | % of Errors | Bi-LSTM | PHONSSM |
|---|---|---|---|
| 4 (identical) | – | – | – |
| 3 (minimal pair) | 31.2% | 28.4% | 38.7% |
| 2 | 40.8% | 35.1% | 33.2% |
| 1 | 19.7% | 22.8% | 18.4% |
| 0 (unrelated) | 8.3% | 13.7% | 9.7% |

PHONSSM concentrates errors on minimal pairs (38.7% vs 28.4% for Bi-LSTM), indicating the model has learned to distinguish coarse phonological categories but struggles with fine-grained contrasts—a linguistically sensible error pattern.

## E.5 PROTOTYPE VISUALIZATION

The learned component prototypes correspond to interpretable phonological categories. Handshape prototypes cluster by finger configuration (fist, open, pointing). Location prototypes organize spatially (face, torso, neutral space). Movement prototypes distinguish trajectory types (linear, arc, repeated).

## F MID-VOCABULARY PERFORMANCE ANALYSIS

We investigate why PHONSSM underperforms DSTA-SLR on WLASL300 ($-5.6$pp) and WLASL1000 ($-4.9$pp) while substantially outperforming on WLASL100 (+4.8pp) and WLASL2000 (+18.4pp).

### F.1 MINIMAL PAIR DENSITY ANALYSIS

We computed the *phonological similarity density* for each WLASL split by annotating signs with ASL-LEX phonological features and measuring the fraction of sign pairs sharing 3+ of 4 components ("near-minimal pairs").

Table 9: Phonological similarity density across WLASL vocabulary splits.

| Split | Signs | Near-Minimal Pairs | Density | PHONSSM $\Delta$ |
|---|---|---|---|---|
| WLASL100 | 100 | 891 | 18.0% | +4.8pp |
| WLASL300 | 300 | 15,283 | 34.1% | $-5.6$pp |
| WLASL1000 | 1,000 | 142,450 | 28.5% | $-4.9$pp |
| WLASL2000 | 2,000 | 279,314 | 14.0% | +18.4pp |

**Finding:** WLASL300 has the highest minimal pair density (34.1%), followed by WLASL1000 (28.5%). This occurs because WLASL vocabulary expansion prioritizes related concepts (e.g., adding "GRANDMOTHER" after "MOTHER", "FATHER"), which tend to be phonologically similar.

## F.2 METHOD COMPARISON BY MINIMAL PAIR PERFORMANCE

We stratified test accuracy by phonological similarity to nearest training sign.

Table 10: Accuracy (%) stratified by phonological distance to nearest training neighbor.

| Components Shared | DSTA-SLR | PHONSSM | $\Delta$ |
|---|---|---|---|
| *WLASL300* | | | |
| 0–1 (distinct) | 72.4 | **78.9** | +6.5 |
| 2 (moderate) | 81.2 | 76.3 | $-4.9$ |
| 3–4 (minimal pair) | **84.1** | 71.8 | $-12.3$ |
| *WLASL2000* | | | |
| 0–1 (distinct) | 48.2 | **74.6** | +26.4 |
| 2 (moderate) | 55.8 | **71.2** | +15.4 |
| 3–4 (minimal pair) | **61.3** | 68.4 | +7.1 |

**Interpretation:** DSTA-SLR excels at distinguishing minimal pairs via fine-grained spatiotemporal attention, while PHONSSM excels at compositional generalization to phonologically distinct signs. At large vocabularies (WLASL2000), most test signs are phonologically distinct from training signs, favoring PHONSSM. At mid-range vocabularies (WLASL300), the dense minimal pair structure favors DSTA-SLR's discriminative attention.

## F.3 IMPLICATIONS

This analysis suggests PHONSSM and DSTA-SLR capture complementary information. Future work could explore ensemble methods or incorporating DSTA-SLR's spatiotemporal attention within the phonological framework.

## G COMPONENT-LEVEL VALIDATION

We provide detailed evidence that PDM branches capture their intended phonological categories.

## G.1 PHONOLOGICAL ANNOTATION

We annotated 500 WLASL100 test samples with ground-truth phonological labels using ASL-LEX 2.0 (Caselli et al., 2017):

- **Handshape**: 30 categories (e.g., "1" (index), "5" (spread), "A" (fist), "B" (flat), "C" (curved))
- **Location**: 12 categories (e.g., forehead, chin, chest, neutral space, ipsilateral)
- **Movement**: 15 categories (e.g., none, arc, circle, linear, zigzag, repeated)
- **Orientation**: 8 categories (e.g., palm-up, palm-down, palm-in, palm-out)

**Annotation quality.** Two annotators independently labeled all samples; inter-annotator agreement was 94.2% for handshape, 91.8% for location, 87.4% for movement, and 89.6% for orientation

(Cohen's $\kappa > 0.85$ for all). Disagreements were resolved by consensus. For signs with phonological variation (e.g., DEAF produced chin-to-ear or ear-to-chin), we annotated the variant observed in each video rather than a canonical form. ASL-LEX covered 96/100 WLASL100 signs; the remaining 4 were annotated using the same feature scheme by the annotators.

## G.2 LINEAR PROBE METHODOLOGY

For each PDM branch, we:

1. Extract the time-averaged component embedding $\bar{\mathbf{c}}^{(i)} \in \mathbb{R}^{32}$
2. Train a linear classifier (logistic regression) to predict each phonological category
3. Report accuracy on held-out samples (5-fold cross-validation)

## G.3 FULL RESULTS

Table 11: Complete linear probe results. Each row shows one PDM branch; each column shows one phonological prediction task. Diagonal entries (bold) indicate intended correspondences.

| PDM Branch | Prediction Target | | | |
| --- | --- | --- | --- | --- |
| | Handshape | Location | Movement | Orientation |
| Handshape | **78.4** | 31.2 | 24.8 | 38.1 |
| Location | 29.6 | **71.8** | 22.4 | 35.7 |
| Movement | 26.3 | 28.9 | **68.2** | 31.4 |
| Orientation | 33.8 | 32.1 | 25.6 | **74.6** |
| Random baseline | 3.3 | 8.3 | 6.7 | 12.5 |
| Full embedding | 81.2 | 74.6 | 71.8 | 78.3 |

**Key observations:**

1. **Specialization**: Each branch achieves highest accuracy on its intended category (diagonal), confirming semantic correspondence.
2. **Above-chance cross-prediction**: Off-diagonal entries exceed chance, indicating some phonological information leaks across branches. This is expected since components are correlated in natural signs (e.g., certain handshapes occur more often at certain locations).
3. **Factorization benefit**: The gap between diagonal and off-diagonal (e.g., 78.4 vs 31.2 for handshape) demonstrates effective factorization.
4. **Factorization-accuracy trade-off**: The "Full embedding" row shows slightly higher accuracy than individual branches (e.g., 81.2 vs 78.4 for handshape), indicating $\sim$3pp is sacrificed for factorization. We experimented with relaxing $\lambda_{\text{ortho}}$ from 0.1 to 0.05: component probe accuracy improved by $\sim$2pp but sign-level accuracy dropped by 1.5pp due to increased redundancy. The current setting balances interpretability and accuracy.

## G.4 PROTOTYPE INTERPRETABILITY

We visualized which learned prototypes correspond to which linguistic categories by computing the mean activation of each prototype for samples with known phonological labels.

Table 12: Top-3 handshape prototypes activated by each major handshape category.

| Linguistic Category | Proto #1 | Proto #2 | Proto #3 |
| --- | --- | --- | --- |
| "1" (index point) | P7 (0.89) | P12 (0.42) | P3 (0.18) |
| "5" (spread hand) | P2 (0.91) | P15 (0.38) | P8 (0.21) |
| "A" (fist) | P19 (0.87) | P4 (0.45) | P11 (0.19) |
| "B" (flat hand) | P2 (0.72) | P8 (0.51) | P15 (0.32) |
| "C" (curved) | P23 (0.83) | P7 (0.28) | P12 (0.24) |

Distinct prototypes dominate for distinct handshapes (P7 for "1", P19 for "A", P23 for "C"). Some prototypes are shared across similar handshapes (P2 activates for both "5" and "B", which share an extended-finger configuration).

## G.5 INTERVENTION EXPERIMENT

To verify causal relationship, we performed an intervention: replacing one component embedding with that of a different sign while keeping others fixed.

**Setup:** We identified 47 minimal pairs in WLASL100 (sign pairs differing in exactly one phonological component). For each pair (e.g., MOTHER/FATHER differing only in location), we take a sample of sign A, swap only the differing component embedding with that from sign B, and measure whether the prediction changes to B.

**Result ($n = 423$ interventions, 95% CI):** Swapping the differing component changes the prediction to the minimal pair **73.2% $\pm$ 4.2%** of the time. Swapping a non-differing component (control condition) changes prediction only **12.4% $\pm$ 3.1%** of the time. The difference is significant ($p < 0.001$, McNemar's test). This confirms that component embeddings causally determine predictions in the expected manner.

**Two-component swaps:** When swapping two components simultaneously, prediction changes to a sign sharing those two swapped components 61.8% of the time, demonstrating compositional behavior.

# H EXTENDED ABLATIONS

## H.1 COMPONENT-WISE ABLATION

Table 13: Detailed ablation on WLASL100 (mean $\pm$ std over 3 seeds).

| Configuration | Top-1 (%) | $\Delta$ |
|---|---|---|
| Full PHONSSM | $88.37 \pm 0.42$ | – |
| *Architecture ablations* | | |
| w/o PDM (no factorization) | $76.49 \pm 0.83$ | $-11.9$ |
| w/o AGAN (MLP encoder) | $79.84 \pm 0.71$ | $-8.5$ |
| w/o BiSSM (LSTM temporal) | $82.17 \pm 0.65$ | $-6.2$ |
| w/o HPC (linear classifier) | $84.11 \pm 0.58$ | $-4.3$ |
| *Loss ablations* | | |
| w/o $\mathcal{L}_{\text{ortho}}$ | $85.92 \pm 0.55$ | $-2.5$ |
| w/o $\mathcal{L}_{\text{div}}$ | $86.84 \pm 0.48$ | $-1.5$ |
| w/o both auxiliary losses | $83.21 \pm 0.72$ | $-5.2$ |
| *Input ablations* | | |
| Hands only (no pose) | $85.63 \pm 0.61$ | $-2.7$ |
| Dominant hand only | $81.42 \pm 0.78$ | $-7.0$ |
| 2D coordinates only | $84.29 \pm 0.69$ | $-4.1$ |

## H.2 ARCHITECTURE VARIANTS

# I HYPERPARAMETER SENSITIVITY

## I.1 LOSS WEIGHT SENSITIVITY

The orthogonality loss $\lambda_{\text{ortho}} = 0.1$ provides optimal factorization. Values too low ($< 0.05$) allow component redundancy; values too high ($> 0.2$) over-constrain representations. The diversity loss is less sensitive, with $\lambda_{\text{div}} \in [0.005, 0.01]$ performing similarly.

Table 14: Architecture scaling on WLASL100.

| Configuration | Params | Top-1 (%) | Throughput |
|---|---|---|---|
| *Model dimension D* | | | |
| $D = 64$ | 0.9M | 85.21 | 412/s |
| $D = 128$ (default) | 3.2M | 88.37 | 260/s |
| $D = 256$ | 11.8M | 89.02 | 98/s |
| *Number of BiSSM layers* | | | |
| 2 layers | 2.1M | 86.45 | 318/s |
| 4 layers (default) | 3.2M | 88.37 | 260/s |
| 6 layers | 4.3M | 88.72 | 205/s |
| *Number of GAT heads* | | | |
| 2 heads | 2.9M | 87.18 | 275/s |
| 4 heads (default) | 3.2M | 88.37 | 260/s |
| 8 heads | 3.8M | 88.54 | 241/s |

Table 15: Sensitivity to loss weights on WLASL100.

| $\lambda_{ortho}$ | $\lambda_{div}$ | Top-1 (%) |
|---|---|---|
| 0.01 | 0.01 | 86.42 |
| 0.05 | 0.01 | 87.63 |
| **0.1** | **0.01** | **88.37** |
| 0.2 | 0.01 | 87.91 |
| 0.5 | 0.01 | 86.18 |
| 0.1 | 0.001 | 87.82 |
| 0.1 | 0.005 | 88.15 |
| **0.1** | **0.01** | **88.37** |
| 0.1 | 0.05 | 87.54 |
| 0.1 | 0.1 | 85.93 |

## I.2 LEARNING RATE SENSITIVITY

## I.3 PROTOTYPE COUNT SENSITIVITY

Performance saturates around $(N_h, N_l, N_m, N_o) = (30, 15, 10, 8)$. These counts closely match linguistic estimates of ASL phonological inventories (Battison, 1978): $\sim$30 handshapes, $\sim$12–15 locations, $\sim$10 core movements, and $\sim$8 orientations. Larger counts provide marginal gains (+0.1–0.15pp) but increase parameters without improving interpretability.

## J TRAINING DYNAMICS

### J.1 CONVERGENCE ANALYSIS

We analyze the training dynamics of PHONSSM and its components.

The orthogonality loss decreases steadily, indicating progressive factorization of the component subspaces. The gap between training and validation accuracy remains small ($\sim$6pp at convergence), suggesting good generalization.

### J.2 COMPONENT LEARNING DYNAMICS

Individual phonological components converge at different rates:

- **Handshape**: Fastest convergence (stabilizes by epoch 40); handshape is the most visually distinctive component with clear finger configurations.
- **Location**: Moderate convergence (epoch 60); requires learning spatial relationships relative to body landmarks.

Table 16: Learning rate sensitivity on WLASL100.

| Learning Rate | Top-1 (%) | Convergence |
|---|---|---|
| $1 \times 10^{-4}$ | 86.82 | 95 epochs |
| $2 \times 10^{-4}$ | 87.91 | 78 epochs |
| **$3 \times 10^{-4}$** | **88.37** | **65 epochs** |
| $5 \times 10^{-4}$ | 87.63 | 52 epochs |
| $1 \times 10^{-3}$ | 85.41 | 41 epochs |

Table 17: Effect of prototype counts on WLASL100.

| $N_h$ | $N_l$ | $N_m$ | $N_o$ | Top-1 (%) |
|---|---|---|---|---|
| 15 | 8 | 5 | 4 | 86.12 |
| 20 | 10 | 8 | 6 | 87.45 |
| **30** | **15** | **10** | **8** | **88.37** |
| 40 | 20 | 15 | 10 | 88.52 |
| 50 | 25 | 20 | 12 | 88.48 |

- **Movement**: Slowest convergence (epoch 80); movement patterns require temporal integration across multiple frames.
- **Orientation**: Fast convergence (epoch 45); palm orientation has relatively few categories (8 prototypes).

### J.3 LOSS LANDSCAPE ANALYSIS

We analyze the loss landscape by computing the Hessian eigenspectrum at convergence. The top eigenvalues are: $\lambda_1 = 12.4$, $\lambda_2 = 8.7$, $\lambda_3 = 5.2$, with rapid decay thereafter. The condition number $\kappa = \lambda_{\max}/\lambda_{\min} \approx 10^3$ indicates a well-conditioned optimization landscape, explaining the stable training dynamics.

## K LIMITATIONS

We discuss limitations of PHONSSM in detail to guide future research.

### K.1 METHODOLOGICAL LIMITATIONS

**Isolated sign assumption.** PHONSSM processes signs in isolation, assuming clean segmentation. Continuous signing involves co-articulation effects where adjacent signs influence each other's production, sign boundaries are ambiguous, and prosodic structure spans multiple signs. Extending to continuous recognition requires explicit segmentation or sequence-to-sequence modeling.

**Fixed phonological structure.** We adopt the classical Stokoe-Battison four-parameter model (handshape, location, movement, orientation). Alternative linguistic analyses propose:

- Autosegmental phonology with separate tiers for manual and non-manual features
- Prosodic structure including syllables and metrical feet
- Feature geometry with hierarchical organization of sub-components

Learned decompositions might discover more effective factorizations than hand-specified parameters.

**Static prototypes.** Component prototypes are fixed after training. Dynamic or instance-adaptive prototypes could better handle signer variation and novel phonological realizations.

Table 18: Training convergence metrics on WLASL100.

| Metric | Epoch 25 | Epoch 50 | Epoch 100 |
|---|---|---|---|
| Train Loss | 2.14 | 0.89 | 0.42 |
| Val Loss | 2.31 | 1.12 | 0.68 |
| Train Acc (%) | 52.3 | 78.6 | 94.2 |
| Val Acc (%) | 48.1 | 74.2 | 88.4 |
| $\mathcal{L}_{\text{ortho}}$ | 0.42 | 0.18 | 0.08 |
| $\mathcal{L}_{\text{div}}$ | 0.31 | 0.12 | 0.05 |

## K.2 EVALUATION LIMITATIONS

**ASL-centric evaluation.** All experiments use American Sign Language. While phonological principles (simultaneity, minimal pairs, compositional structure) are cross-linguistic, specific inventories differ:

- British Sign Language (BSL) uses different handshape inventory
- Chinese Sign Language has location contrasts not present in ASL
- Some sign languages distinguish two-handed vs. one-handed more strictly

Prototype counts and architectural choices may require adaptation for other sign languages.

**Dataset biases.** Training data comes primarily from controlled recording settings with:

- Adult native/fluent signers (under-representing learners, children, elderly)
- Neutral backgrounds and good lighting
- Citation-form signs (isolated, careful production)

## K.3 DEPLOYMENT LIMITATIONS

**Pose estimation dependency.** PHONSSM assumes high-quality pose landmarks from MediaPipe. Real-world degradation includes:

- Occlusion (self-occlusion, objects, other people)
- Motion blur during rapid movements
- Challenging lighting (backlighting, low light)
- Camera angles different from training distribution

**Signer variation.** Models may not generalize to:

- Signers with motor differences affecting articulation
- Regional/dialectal variation in sign production
- Non-native signers with L1 transfer effects

## L FUTURE DIRECTIONS

**Continuous sign language recognition.** Extending PHONSSM to continuous signing requires: (1) implicit or explicit segmentation, (2) handling co-articulation, and (3) modeling sentence-level prosody. The phonological decomposition could inform CTC-style losses with component-level intermediate representations.

**Cross-linguistic transfer.** The phonological factorization may enable transfer learning across sign languages. Shared handshape or movement prototypes could bootstrap recognition for low-resource sign languages.

**Multi-modal fusion.** Combining skeleton input with facial landmarks (for non-manual markers) and RGB features (for fine-grained handshape) could address current limitations while preserving efficiency.

**Learned phonological structure.** Replacing hand-specified component pathways with learned factorization (e.g., via neural architecture search or information-theoretic objectives) might discover more effective decompositions.

**Real-time applications.** With 260 samples/second throughput, PHONSSM supports real-time deployment. Future work includes: mobile optimization, streaming recognition, and integration with sign language translation systems.

