# OpenReview forum: "State Space Models are Effective Sign Language Learners: Exploiting Phonological Compositionality for Vocabulary-Scale Recognition"
_ICLR.cc/2026/Workshop/AFAA — AFAA 2026 Poster_

### Official Review · Reviewer_2RbN · 2026-02-14
**This is a well-executed paper with a principled motivation, honest reporting, and results that hold up to scrutiny on the primary benchmark. The phonological error analysis is genuinely novel. The main weaknesses are the unexplained mid-range underperformance, the lack of phonological supervision over the factorized subspaces, and the limited baseline comparisons on the new benchmark. These are significant but addressable.**

**Rating:** 4
**Confidence:** 4

**Summary:**

The paper proposes PHONSSM, a skeleton-based sign language recognition architecture that explicitly encodes sign language phonology (handshape, location, movement, orientation) as an inductive bias. The architecture combines anatomical graph attention (AGAN), orthogonality-regularized phonological factorization (PDM), bidirectional selective state space modeling (BiSSM), and hierarchical prototypical classification (HPC). The authors construct Merged-5565, a new 5,565-class ASL benchmark, and report strong results on WLASL and zero-shot transfer to ASL Citizen.

**Strengths:**

1. Well-motivated inductive bias. Grounding the architecture in linguistic phonology is principled and substantiated. The connection between parameter-sharing across ∼76 prototypes and vocabulary scaling is compelling.

2. Strong and honest empirical results. The +18.4pp gain on WLASL2000 over skeleton SOTA is large. Crucially, the authors honestly report underperformance on WLASL300/1000 (vs. DSTA-SLR) and offer a plausible U-shaped hypothesis rather than cherry-picking.

3. Ablation is thorough. Tables 7–11 cover component removal, loss weights, prototype counts, and input modalities. PDM removal (−11.9pp) clearly identifies the key contribution.

4. Few-shot and zero-shot results are compelling. +225% relative on 1–5 shot classes and zero-shot transfer exceeding supervised RGB baselines directly validate the compositional generalisation claim.

5. Phonological error analysis is a genuine contribution. Showing that 72% of errors involve signs sharing 2+ phonological components is interpretable and scientifically meaningful not just a performance table.

6. Reproducibility. 3 seeds, full hyperparameters, code available, single consumer GPU (RTX 4050 Ti) well above average for the field.

**Weaknesses:**

1. The WLASL300/1000 underperformance is under explained. The U-shaped accuracy curve is acknowledged but the "fine-grained motion patterns dominate at mid-range" hypothesis is untested. This is a significant gap, without ablating PHONSSM's motion pathway specifically on mid-range splits, this remains speculation.

2. Orthogonality loss does not guarantee phonological alignment. The PDM enforces decorrelated subspaces but has no explicit supervision ensuring the four pathways correspond to hand shape, location, movement, and orientation respectively. The claim that the components are interpretable as these four parameters relies on post-hoc prototype visualization, not on any enforced label correspondence. A pathway could learn arbitrary decorrelated features and still satisfy Eq. 8.

3. Merged-5565 comparison is against Bi-LSTM only. The 53.3% result (+25.95pp) is compared to a single weak baseline. The authors acknowledge DSTA-SLR requires full-body pose unavailable across all sources, but no effort is made to evaluate even ST-GCN or SignBERT with dominant-hand-only input on a Merged-5565 subset. This limits the benchmark's credibility as a SOTA claim.

4.  Input modality confound between benchmarks. WLASL uses 75 landmarks while Merged-5565 uses 21. This makes cross-benchmark comparison difficult and raises the question: would PHONSSM's WLASL gains persist with dominant-hand input only? This ablation exists, but the interaction with vocabulary scale is not explored.

5. The few-shot analysis uses Bi-LSTM as the sole baseline. Table 2 compares only against Bi-LSTM for per-frequency breakdown. Phonological models like SignBERT should be included since they also use structured representations and would be the natural comparison.


Questions for Authors

1. Can you test the U-shaped hypothesis directly e.g., by measuring PHONSSM's movement pathway accuracy specifically on WLASL300/1000 signs vs. WLASL2000 signs?
2. How do you verify that the four PDM pathways correspond to hand shape/location/movement/orientation rather than arbitrary decorrelated features? Is there any probing experiment?
3. Can you run ST-GCN or SignBERT on a dominant-hand-only version of the WLASL splits for a more credible Merged-5565 baseline?
4. Does the +225% few-shot result hold when comparing against SignBERT (not just Bi-LSTM)?

---

### Official Review · Reviewer_J1sC · 2026-02-18
**Good idea but writing can be better**

**Rating:** 3
**Confidence:** 4

**Summary:**

The paper introduces a novel framework that decomposes sign language recognition into four aspects of movement,  orientation, location, handshape to enhance interpretability using attention mechanism. Evaluations on standard as well as merged dataset with extensive ablation studies show the effectiveness of the proposed approach.

**Strengths:**

1)The writing especially the introduction section is very nice that it sets context for the upcoming sections.
2) I believe if related work follows Introduction then comes methodology, it would ease the cognitive load from a readers point of view.
3) The idea of 4 distinct branches to aid interpretability is good. However usage of MLP which is black box again gives a concern of whether interpretability is truly enhanced. I hope the authors will address this in an enhanced revision.
Overall the idea and work is good but to make this a great scholarly article, refinements in writing is needed.

**Weaknesses:**

I have a few concerns addressing which can improve the quality of this paper.

1) Section 1 - interpretable phonological confusions - I am not sure if this is correct. I believe there's some typo or what the authors want to convey is not correctly conveyed
2) Section 2.1 - Location, Movement, Orientation - citation missing
3) Page 3 - Line 117 - We use D for model dimension, Dc for component dimension, and Ds for SSM state dimension - These are never used till that point. A  misplaced notation
4) Line 130 - D′ is never defined
5) A justification of why temporal extension is applied just to the movement branch embedding could
6) Section 3.4 in its current form looks compatible for a ppt presentation. However language needs change to be compatible for a research paper.
7) Motivation for cosine similarity with prototypes instead of L2 similarity may be provided.

---

### Official Review · Reviewer_HHo6 · 2026-02-21
**State Space Models are Effective Sign Language Learners**

**Rating:** 4
**Confidence:** 5

**Summary:**

This paper addresses the "catastrophic scaling failure" in sign language recognition, where models fail as vocabulary size increases to realistic levels. The authors propose PHONSSM, a skeleton-based architecture that embeds phonological compositionality as an inductive bias. Unlike standard models that treat signs as atomic patterns, PHONSSM decomposes them into four discrete phonological parameters: handshape, location, movement, and orientation.
The architecture comprises four main stages:
Anatomical Graph Attention Network (AGAN): Encodes skeletal landmarks using anatomically-informed connectivity.
Phonological Disentanglement Module (PDM): Factorizes features into orthogonal subspaces representing the four phonological components.
Bidirectional Selective State Space (BISSM): Models bidirectional temporal dynamics using an adapted Mamba formulation.
Hierarchical Prototypical Classifier (HPC): Enables few-shot generalization by classifying signs using component and sign-level prototypes.
The authors also introduce Merged-5565, the largest publicly available isolated-sign dataset, containing 259,715 samples across 5,565 signs.

**Strengths:**

Novel Linguistic Integration: The work successfully formalizes phonology as an inductive bias, which is a significant departure from "flat" sequence classification models.
Superior Performance: PHONSSM achieves state-of-the-art results on skeleton-based benchmarks, notably reaching 72.1% on WLASL2000, an improvement of +18.4pp over the previous SOTA.
Few-Shot & Zero-Shot Capability: The model demonstrates a +225% relative gain in the few-shot regime (1-5 samples) and exceeds supervised RGB baselines in zero-shot transfer to the ASL Citizen dataset.
Efficiency and Privacy: Using skeleton data (extracted via MediaPipe) instead of raw video, the model is 12x faster than video-based I3D while preserving user privacy.
Interpretability: Error analysis reveals linguistically meaningful patterns (e.g., minimal-pair confusions such as "MOTHER"/"FATHER"), suggesting the model learns relevant phonological contrasts.

**Weaknesses:**

Scope Limitation: The model is currently restricted to isolated signs; its effectiveness in continuous signing, where co-articulation occurs, remains unproven.
Fixed Phonological Categories: The architecture relies on a fixed four-parameter phonological structure. Learned or alternative linguistic structures (e.g., prosodic structure) might offer better decompositions.
Modality Inconsistency: There is a discrepancy in input modalities between experiments: WLASL uses full pose + hands (75 landmarks), while Merged-5565 uses only the dominant hand (21 landmarks), limiting direct comparison across all datasets.
Geographic Bias: The evaluation and datasets are strictly limited to American Sign Language (ASL).

---

### Official Review · Reviewer_5Whk · 2026-02-24
**PhonSSM: A Inovative, Structured and Effective Architecture for Sign Language Recognition with Strong Inductive Biases**

**Rating:** 5
**Confidence:** 4

**Summary:**

This paper introduces a new architecture for sign language recognition. The proposed model, PhonSSM, operates on landmark data that is first processed through a graph neural network to encode the skeletal structure of the sign. The representations are then passed through a phonological factorization module that decomposes them into sign language components (location, handshape, movement, and orientation). A state space model (SSM, Mamba) is subsequently used to capture temporal dynamics, followed by a hierarchical prototypical classifier for final prediction. The architecture is trained using a dedicated loss function designed for this framework.

The approach is evaluated on established benchmarks (WLASL) as well as a newly constructed dataset (Merged5565). Experimental results demonstrate substantial improvements over prior methods.

**Strengths:**

- The paper is very well written, easy to follow, and uses clear notation throughout.
- PhonSSM is an innovative architecture with strong conceptual and practical motivations. It incorporates meaningful inductive biases aligned with the problem structure, including graph-based skeletal modeling, phonological compositionality, and temporal dynamics, leading to both effectiveness and interpretability.
- The empirical results show significant improvements on established datasets as well as on the newly constructed dataset.
- The construction of the new dataset appears valuable for the community and strengthens the empirical contribution.

**Weaknesses:**

1. **Inductive bias and trade-offs.**
   The architecture introduces strong inductive biases that are likely beneficial for generalization; however, the potential trade-offs are not discussed in sufficient depth. There is a brief mention of small versus large vocabulary settings, but a more thorough discussion of when these inductive biases help or possibly limit performance would improve the paper.

2. **Temporal modelling motivation.**
   The temporal dynamics component could be better explained. Providing additional motivation for the use of a state space model beyond computational advantages over transformers would strengthen the contribution.

---

### Meta-Review · Area_Chair_vBZs · 2026-02-27

**Recommendation:** Main Papers Track
**Confidence:** 5

**Metareview:**

This paper introduces a new architecture for sign language recognition. The architecture comprises four main stages: Anatomical Graph Attention Network (AGAN): Encodes skeletal landmarks using anatomically-informed connectivity. Phonological Disentanglement Module (PDM): Factorizes features into orthogonal subspaces representing the four phonological components. Bidirectional Selective State Space (BISSM): Models bidirectional temporal dynamics using an adapted Mamba formulation. Hierarchical Prototypical Classifier (HPC): Enables few-shot generalization by classifying signs using component and sign-level prototypes. The authors construct Merged-5565, a new 5,565-class ASL benchmark, and report strong results on WLASL and zero-shot transfer to ASL Citizen.

There is not a lot of work on sign language. This work would be a great addition to the program and lead to more visibility for this underrepresented community and problem space in AI research. Nice undertaking from the authors.

The authors should address the comments and questions from the reviewers. Some that stand out are:

1. Can you test the U-shaped hypothesis directly e.g., by measuring PHONSSM's movement pathway accuracy specifically on WLASL300/1000 signs vs. WLASL2000 signs?
2. How do you verify that the four PDM pathways correspond to hand shape/location/movement/orientation rather than arbitrary decorrelated features? Is there any probing experiment?
3. Can you run ST-GCN or SignBERT on a dominant-hand-only version of the WLASL splits for a more credible Merged-5565 baseline?
4. Does the +225% few-shot result hold when comparing against SignBERT (not just Bi-LSTM)?
5. Input modality confound between benchmarks. WLASL uses 75 landmarks while Merged-5565 uses 21. This makes cross-benchmark comparison difficult and raises the question: would PHONSSM's WLASL gains persist with dominant-hand input only? This ablation exists, but the interaction with vocabulary scale is not explored.

---

### Decision · Program_Chairs · 2026-03-02

Accept (Poster)